# Delay Minimization in RIS-Assisted URLLC Systems under Reliability Constraints

**DOI:** 10.3390/e25060857

**Published:** 2023-05-26

**Authors:** Xiaoyang Song, Yingxin Zhao, Wannan Zhao, Hong Wu, Zhiyang Liu

**Affiliations:** 1College of Electronic Information and Optical Engineering, Nankai University, Tianjin 300350, China; 2120210336@mail.nankai.edu.cn (X.S.);; 2Tianjin Key Laboratory of Optoelectronic Sensor and Sensing Network Technology, Tianjin 300350, China

**Keywords:** URLLC, RIS, ADMM, IoT network

## Abstract

The ultra-reliable and low-latency communication (URLLC) systems are expected to support the stringent quality of service (QoS) demands in the Internet of Things (IoT) networks. In order to support the strict latency and reliability constraints, it is preferable to deploy a reconfigurable intelligent surface (RIS) in the URLLC systems to improve the link quality. In this paper, we focus on the uplink of an RIS-assisted URLLC system, and we propose to minimize the transmission latency under the reliability constraints. To solve the non-convex problem, a low-complexity algorithm is proposed by using the Alternating Direction Method of Multipliers (ADMM) technique. The RIS phase shifts optimization, which is typically non-convex, is efficiently solved by formulating as a Quadratically Constrained Quadratic Programming (QCQP) problem. Simulation results verify that our proposed ADMM-based method is able to achieve better performance than the conventional semi-definite relaxation (SDR)-based method with lower computational complexity. Our proposed RIS-assisted URLLC system is able to significantly reduce the transmission latency, which highlights the great potential in deploying RIS in the IoT networks with strict reliability requirements.

## 1. Introduction

Ultra-reliable and low-latency communication (URLLC) has been envisioned as a vital scenario for mobile communication systems to enable Internet of Things (IoT) applications such as industrial automation [1,2] and unmanned aerial vehicle (UAV)-assisted communication [3,4]. Compared to an enhanced mobile broadband (eMBB) scenario which mainly focuses on the spectral efficiency, URLLC has imposed strict requirements on the end-to-end (E2E) transmission latency (e.g., less than 1 ms) and reliability (e.g., decoding error probability (DEP) less than $10^{-9}$). Short packets are, therefore, transmitted in the URLLC systems to support the quality-of-service (QoS) demands. Generally, the maximum achievable rate with a short packet scheme is lower than the conventional communication channels, as the Shannon capacity limit is further penalized by terms related to the blocklength and the error probability bound [5]. To simultaneously satisfy the latency and reliability requirements, high-quality communication links are required, making it difficult to be implemented in the complicated propagation environments.

The reflecting intelligent surface (RIS) has recently been proposed as a promising technology in recent years for future wireless communication to increase the system spectral and energy efficiencies [6,7]. The RIS is a passive meta-material panel composed of a large number of low-cost passive reflecting elements, and it can be used to reconfigure the wireless propagation environments by inducing independent phase shifts on the incident signals. By adaptively tuning the phase shifts of the RIS according to the channel state information (CSI), the reflected signals can be constructively added up at the desired direction, and the communication performance can be thus significantly improved [8,9,10]. As each reflecting element is passive and without a radio frequency chain, the RIS systems cost much lower energy than the conventional relay systems. It is, therefore, expected that the RIS can play a crucial role in meeting the high requirements on latency and reliability in the URLLC systems.

The advantages in utilizing RIS on the URLLC systems have recently sparkled a flurry of research activities [11,12,13]. For instance, Ref. [11] studied the joint optimization of user grouping and the passive beamforming so as to minimize the total blocklength. A multi-objective optimization problem was studied for maximizing the achievable rate while minimizing the utilized channel blocklength in the downlink channel of the multiuser RIS-URLLC systems [12]. The effect of RIS in the device-to-device (D2D) URLLC transmissions was also studied in [13], where a robust active and passive beamforming design was proposed to maximize the number of actuators with successful decoding. Although it is generally acknowledged that the performance of a URLLC system can be significantly improved by the assistance of RIS, most articles focus on the downlink of the RIS-assisted URLLC systems. In an IoT network, it is also important for the base station (BS) to receive the timely information under the high reliability constraint [14,15,16]. However, how to efficiently solve the RIS problem under the reliability constraint remains challenging. Under the short packet scheme, the SINR, which is related to the phase shifts, is incorporated in both the Shannon capacity term and the short packet penalty term, making it even more difficult to find the optimum. As we will show in this paper, the Alternating Direction Method of Multipliers (ADMM) technique can be adopted to decompose the non-convex problem to a series of convex problems.

In particular, we consider the uplink transmission in an RIS-assisted URLLC system with multiple users, and we investigate the transmission scheme that minimizes the total transmission delay of the whole system, which is measured by the total blocklength, under the DEP constraints. As the feasible region is non-convex, the primal problem is converted to several convex problems and solved by using the ADMM technique. Simulation results show that the utilization of the RIS is able to improve the propagation environment and thus significantly reduce the transmission latency under the maximum DEP constraints. Our proposed method is able to achieve better performance than the conventional semi-definite relaxation (SDR)-based method with much lower complexity.

Throughout this paper, C denotes the set of complex numbers. Suppose A, x and *x* are a matrix, a vector and a complex number, respectively. $A^{H}$ and rank(A) denote the conjugate-transpose and the rank of A, respectively. $A_{i,k}$ denotes the element in the *i*-th row and *k*-th column of A. $I_{N}$ denotes the identity matrix with dimension *N*. ${\left[x\right]}_{i}$ is the *i*-th element of x. The diag(x) denotes converting vector x into a diagonal matrix. $∥x∥$ and $∥A∥$ denote the $\ell_{2}$ norm of x and Frobenius norm of A. R{x} denotes the real part of value *x*.

## 2. System Model and Problem Formulation

### System Model

Consider the uplink transmission of an ultra-reliable and low-latency communication (URLLC) system as illustrated in Figure 1, which is composed of a control center (CC) with *N* antennas, an reconfigurable intelligent surface (RIS) with *M* component elements and a set K of single-antenna Internet of Things (IoT) devices with |K|=K. The RIS is equipped with *M* elements, and the phase shifts of the *M* elements are controlled by the CC through an RIS controller. Due to the severe path loss, we assume that the signals that are reflected by the RIS twice or more are negligible. Suppose that perfect CSI is available at the CC through the existing channel estimation technique [17]. By assuming a quasi-static flat-fading channel model, the received signal at the CC from the *k*-th IoT device can be modeled as
(1)$$y_{k}=\sqrt{P_{k}}\cdot h_{k}\cdot s_{k}+n_{0},$$
for k∈K, where $P_{k}$ is the transmission power of device *k* and $s_{k}$ is the transmitted symbol by device *k*, which is modeled as a circular symmetric complex Gaussian (CSCG) random variable with zero mean and unit variance, i.e., $s_{k}∼\mathrm{CN}\left(0,1\right)$. $n_{0}$ is the additive white Gaussian noise (AWGN) with zero mean and variance $\sigma^{2}$. $h_{k}$ represents the cascaded channel between device *k* and the CC, which can be modeled as
(2)$$h_{k}=h_{k}^{IC}+G^{H}\cdot \Phi \cdot f_{k}$$
where $h_{k}^{IC}\in C^{N\times 1}$, $G\in C^{M\times N}$, and $f_{k}\in C^{M\times 1}$ denote the channel gain coefficients between device *k* and the CC, the RIS and the CC, and device *k* and the RIS, respectively. $h_{k}^{IC}$, G and $f_{k}$ are modeled as $h_{k}^{IC}=\sqrt{\xi \cdot {\left(\frac{d_{I_{k}C}}{d_{0}}\right)}^{-\alpha_{I_{k}C}}}{\bar{h}}_{k}^{IC}$, $G=\sqrt{\xi \cdot {\left(\frac{d_{RC}}{d_{0}}\right)}^{-\alpha_{RC}}}\bar{G}$ and $f_{k}=\sqrt{\xi \cdot {\left(\frac{d_{I_{k}R}}{d_{0}}\right)}^{-\alpha_{I_{k}R}}}{\bar{f}}_{k}$ where ξ is the path loss at the reference distance $d_{0}$. $d_{I_{k}C}$, $d_{RC}$ and $d_{I_{k}R}$ represent communication distances between device *k* and the CC, the RIS and the CC, and device *k* and the RIS, respectively. $\alpha_{I_{k}C}$, $\alpha_{RC}$ and $\alpha_{I_{k}R}$ represent path loss exponents between device *k* and the CC, the RIS and the CC, and device *k* and the RIS, respectively. ${\bar{h}}_{k}^{IC}$, $\bar{G}$ and ${\bar{f}}_{k}$ are assumed to be independent Rayleigh fading between IoT device *k* and CC, the RIS and CC, and IoT device *k* and the RIS, respectively. Φ is the reflection coefficient matrix of the RIS. In general, Φ is a diagonal matrix given by Φ = diag(θ) with $\theta ={\left[\theta_{1},\cdots ,\theta_{M}\right]}^{T}$, where $\theta_{m}$ denotes the reflection coefficient at the *m*-th reflecting element, m=1,2,⋯,M. For simplicity, (Equation 2) can be further rewritten as
(3)$$h_{k}=h_{k}^{IC}+Q_{k}\cdot \theta ,$$
where $Q_{k}=G^{H}\cdot \mathrm{diag}\left(f_{k}\right)$. The CC estimates the symbol from device *k* according to the received signal vector $y_{k}$, and the estimated symbol can be obtained as
(4)$${\hat{s}}_{k}=w_{k}^{H}y_{k},$$
where $w_{k}$ is the received beamforming vector at the CC to estimate the symbol from device *k* with $∥w_{k}∥=1$.

Note that in a URLLC scenario, the packet length $D_{k}$ transmitted from device *k* should be rather short due to the strict latency constraint. It is necessary to adopt the coding rate in the finite blocklength regime to measure the maximum achievable rate. Specifically, the maximum achievable rate in terms of bits per channel blocklength of the device *k* in the AWGN channel can be approximately [5] defined as
(5)$$\frac{D_{k}}{m_{k}}=\log_{2}\left(1+\gamma_{k}\right)-\sqrt{\frac{V_{k}}{m_{k}}}\cdot Q^{-1}\left(\epsilon_{k}\right),$$
where $D_{k}$, $m_{k}$ and $\epsilon_{k}$ denote the length of the data packet, the channel blocklength and the decoding error probability (DEP) of device *k*, respectively. $Q^{-1}\left(x\right)$ is the inverse Q-function defined as $Q^{-1}\left(x\right)=\int_{x}^{\infty }e^{-\frac{t^{2}}{2}}dx$. $V_{k}$ is the channel dispersion, which is given as
(6)$$V_{k}=\left(1-\frac{1}{{\left(1+\gamma_{k}\right)}^{2}}\right)\cdot \log_{2}^{2}e.$$

$\gamma_{k}$ is the signal-to-interference-plus-noise ratio (SINR) for the device *k* at CC, which can be characterized as
(7)$$\gamma_{k}=\frac{P_{k}{\left|w_{k}^{H}h_{k}\right|}^{2}}{\sum_{i=1,i\neq k}^{K}P_{i}{\left|w_{k}^{H}h_{i}\right|}^{2}+\sigma^{2}}.$$

According to (Equation 5), the DEP of the *k*-th device can be expressed as a function of blocklength $m_{k}$, beamforming vector $w_{k}$ and the RIS phase shift θ, i.e.,
(8)$$\begin{array}{cc} \epsilon_{k}\left(m_{k},\gamma_{k}\right) & =Q(\frac{m_{k}\log_{2}\left(1+\gamma_{k}\right)-D_{k}}{\sqrt{m_{k}V_{k}}})\\ \; & =Q(f\left(m_{k},\gamma_{k}\right)). \end{array}$$
where $f(m_{k},\gamma_{k})$ is defined as
(9)$$f\left(m_{k},\gamma_{k}\right)≜\frac{m_{k}\log_{2}\left(1+\gamma_{k}\right)-D_{k}}{\sqrt{m_{k}V_{k}}}.$$

Our goal is to minimize the total blocklength of the system by jointly optimizing the phase shifts of the RIS elements, the beamforming vector, and the blocklength of each user, subject to the practical constraints on the devices’ maximum DEP, and the unit modulus constraints of the RIS element and beamforming vector. By defining $m≜[m_{1},\cdots ,m_{K}]$ as the blocklength vector of all IoT devices, the total blocklength minimization problem can be formulated as
$$\begin{array}{cc} (P1): & \underset{m,\{w_{k}\},\theta }{\mathrm{minimize}}\sum_{k=1}^{K}m_{k} \end{array}$$
(10)$$\begin{array}{cc} s.t.\; & \epsilon_{k}\left(m_{k},w_{k},\theta \right)\leq \epsilon_{max},\forall k\in K, \end{array}$$
(11)$$\begin{array}{cc} & ∥w_{k}∥=1,\forall k\in K,\;\;\;\;\;\;\;\;\;\;\; \end{array}$$
(12)$$\begin{array}{c} |\theta_{m}|=1,\forall m,\;\;\;\;\;\;\;\;\;\;\;\;\;\;\;\;\; \end{array}$$
where $\epsilon_{max}$ is the maximum DEP. The constraint (Equation 10) denotes the reliability requirement in the URLLC scenario. The (Equation 11) and (Equation 12) constrained the phase shift of each RIS element and received beamforming vector, respectively. As the feasible region in (P1) is non-convex due to the constraints (Equation 10)–(Equation 12), we propose an efficient iterative algorithm based on the ADMM method [18] to solve (P1).

## 3. Proposed Algorithm

In this section, (P1) is first transformed to a more tractable form, and an efficient ADMM-based algorithm is proposed to jointly optimize the channel blocklength vector m, phase shift of RIS θ and received beamforming vector $\left\{w_{k}\right\}$.

### 3.1. ADMM-Problem Reformulation

We first introduce an auxiliary variable $\varphi \in C^{M\times 1}$ and reformulate (P1) as
(13)$$\begin{array}{cc} (P2): & \underset{m,\{w_{k}\},\theta ,\varphi }{\mathrm{minimize}}\sum_{k=1}^{K}m_{k}\\ s.t.\; & \left(10),(11\right)\\ & |\varphi_{m}|=1,\forall m, \end{array}$$
(14)$$\begin{array}{c} \varphi =\theta . \end{array}\;\;\;\;\;\;\;$$

We introduce an indicator function $I(m,w_{k},\theta ,\varphi )$ to impose the constraints (Equation 10), (Equation 11) and (Equation 13), where $I(m,w_{k},\theta ,\varphi )=0$ if the variables $m,w_{k},\theta ,$ and φ lie in feasible regions, and $I(m,w_{k},\theta ,\varphi )=\infty$ otherwise. (P2) can be then rewritten as
$$\begin{array}{cc} (P3): & \underset{m,\{w_{k}\},\theta ,\varphi }{\mathrm{minimize}}\sum_{k=1}^{K}m_{k}+I\left(m,w_{k},\theta ,\varphi \right)\\ s.t.\; & (14). \end{array}$$

We formulate the augmented Lagrangian (AL) function of (P3) as
(15)$$\begin{array}{c} L\left(m,w_{k},\theta ,\varphi ,\lambda \right)=\sum_{k=1}^{K}m_{k}+I\left(m,w_{k},\theta ,\varphi \right)+\frac{\rho }{2}\cdot {∥\theta -\varphi +\lambda ∥}^{2}. \end{array}$$
where ρ>0 is the penalty parameter, and $\lambda \in C^{M\times 1}$ is the dual variable. The variables $m,w_{k},\theta ,\varphi$ and λ can be iteratively updated by applying the ADMM method.

Before deriving the detailed algorithm, the following proposition and lemma are first introduced.

**Proposition** **1.**
*The DEP $\epsilon_{k}\left(m_{k},\gamma_{k}\right)$ given in (Equation 8) is a strict monotonically decreasing function with respect to $m_{k}$ and $\gamma_{k}$.*


**Proof.** According to Proposition 1 in [19], $\epsilon_{k}\left(m_{k},\gamma_{k}\right)$ is a strict decreasing function with respect to $\gamma_{k}$. To prove the monotonicity of $\epsilon_{k}\left(m_{k},\gamma_{k}\right)$ with respect to $m_{k}$, we derive the partial derivative $\frac{\partial \epsilon_{k}\left(m_{k},\gamma_{k}\right)}{\partial m_{k}}$ as
(16)$$\frac{\partial \epsilon_{k}\left(m_{k},\gamma_{k}\right)}{\partial m_{k}}=-\frac{1}{\sqrt{2\pi }}\cdot e^{-\frac{f^{2}\left(m_{k},\gamma_{k}\right)}{2}}\cdot \left[\frac{\log_{2}\left(1+\gamma_{k}\right)+\frac{D_{k}}{m_{k}}}{2\sqrt{m_{k}V_{k}}}\right].$$By noting that $\frac{\partial \epsilon_{k}\left(m_{k},\gamma_{k}\right)}{\partial m_{k}}<0$ holds for all $m_{k}>0$, the DEP $\epsilon_{k}\left(m_{k},\gamma_{k}\right)$ is a strict decreasing function with respect to $m_{k}$.    □

Based on Proposition 1, the optimal conditions of (P2) can be proved in the following lemma.

**Lemma** **1.**
*The equality in (Equation 10) holds at the optimum.*


**Proof.** We prove it by using contradiction. Denote the blocklength and SINR of device *k* at the optimum of (P2) as ${m_{k}}^{*}$ and ${\gamma_{k}}^{*}$, respectively. If $\epsilon_{k}\left(m_{k}^{*},\gamma_{k}^{*}\right)<\epsilon_{max}$, according to Proposition 1, we can always decrease the blocklength $m_{k}^{*}$ to a $m_{k}^{†}<m_{k}^{*}$ such that $\epsilon_{k}\left(m_{k}^{†},\gamma_{k}^{*}\right)=\epsilon_{max}$ without violating any other constraints. This contradicts the optimality of $m_{k}^{*}$. Therefore, we conclude that $\epsilon_{k}\left(m_{k},\gamma_{k}\right)=\epsilon_{max}$ can always be guaranteed at the optimum.    □

### 3.2. Block Update

#### 3.2.1. Update m

At the (t+1)-th iteration, with given $w_{k}^{\left(t\right)},\theta^{\left(t\right)},\varphi^{\left(t\right)}$ and $\lambda^{\left(t\right)}$, the subproblem that optimizes the blocklength m is given as
(17)$$\begin{array}{cc} (P_{m}): & \underset{m}{\mathrm{minimize}}\sum_{k=1}^{K}m_{k}\\ s.t.\; & \epsilon_{k}\left(m_{k}\right)\leq \epsilon_{max},\forall k\in K. \end{array}$$

According to Lemma 1, the constraint (Equation 17) achieves the equality at the optimum. Therefore, the optimal value of $m_{k}$ can be obtained by solving the following equality according to (Equation 5) as
(18)$$m_{k}\ln\left(1+\gamma_{k}^{\left(t\right)}\right)-Q^{-1}\left(\epsilon_{max}\right)\cdot \sqrt{\left[1-{\left(1+\gamma_{k}^{\left(t\right)}\right)}^{-2}\right]m_{k}}-\ln2D_{k}=0,$$
which is a quadratic equation with respect to $\sqrt{m_{k}}$, and it can be analytically solved as
(19)$$m_{k}^{\left(t+1\right)}={\left(\frac{-b_{k}^{\left(t\right)}+\sqrt{b_{k}^{(t)2}-4a_{k}^{\left(t\right)}c_{k}^{\left(t\right)}}}{2a_{k}^{\left(t\right)}}\right)}^{2},$$
where $a_{k}^{\left(t\right)}=\ln\left(1+\gamma_{k}^{\left(t\right)}\right)$, $b_{k}^{\left(t\right)}=-Q^{-1}\left(\epsilon_{max}\right)\cdot \sqrt{1-{\left(1+\gamma_{k}^{\left(t\right)}\right)}^{-2}}$ and $c_{k}^{\left(t\right)}=-\ln2D_{k}$.

#### 3.2.2. Update $\left\{w_{k}\right\}$

As Lemma 1 shows, the DEP $\epsilon_{k}=\epsilon_{\max}$ when the total blocklength is minimized. In addition, Proposition 1 proves that $\epsilon_{k}$ monotonically decreases with respect to blocklength $m_{k}$ and SINR $\gamma_{k}$. To minimize the total blocklength $\sum_{k=1}^{K}m_{k}$ while ensuring that $\epsilon_{k}=\epsilon_{\max}$, the SINR $\gamma_{k}$ should be maximized. The optimization of $\left\{w_{k}\right\}$ is, therefore, equivalent to finding $\left\{w_{k}\right\}$ that maximizes $\gamma_{k}$. In particular, at the (t+1)-th iteration, for given $m^{\left(t+1\right)},\theta^{\left(t\right)},\varphi^{\left(t\right)}$ and $\lambda^{\left(t\right)}$, the optimal $w_{k}$ can be obtained as
(20)$$w_{k}^{\left(t+1\right)}=\underset{∥w_{k}∥=1}{\mathrm{argmax}}\frac{P_{k}w_{k}^{H}h_{k}^{\left(t\right)}h_{k}^{(t)H}w_{k}}{w_{k}^{H}\left(\sum_{i\neq k}^{K}P_{i}h_{i}^{\left(t\right)}h_{i}^{(t)H}+\sigma^{2}I_{N}\right)w_{k}},$$
which can be analytically solved as
(21)$$w_{k}^{\left(t+1\right)}=\frac{{\left(\sigma^{2}I_{N}+\sum_{i\neq k}^{K}P_{i}h_{i}^{\left(t\right)}h_{i}^{(t)H}\right)}^{-1}h_{k}^{\left(t\right)}}{∥{\left(\sigma^{2}I_{N}+\sum_{i\neq k}^{K}P_{i}h_{i}^{\left(t\right)}h_{i}^{(t)H}\right)}^{-1}h_{k}^{\left(t\right)}∥}$$

#### 3.2.3. Update
θ


For given $m^{\left(t+1\right)},w_{k}^{\left(t+1\right)},\varphi^{\left(t\right)}$ and $\lambda^{\left(t\right)}$, the optimal phase shifts θ can be obtained by solving
(22)$$\begin{array}{cc} (P_{\theta }): & \underset{\theta }{\mathrm{minimize}}{∥\theta -\varphi^{\left(t\right)}+\lambda^{\left(t\right)}∥}^{2}\\ s.t.\; & \epsilon_{k}\left(\theta \right)\leq \epsilon_{max},\forall k\in K. \end{array}$$

According to Proposition 1, we first convert the constraint (Equation 22) to an equivalent SINR form, i.e.,
(23)$$\gamma_{k}\geq \gamma_{k,th},$$
where $\gamma_{k,th}$ can be computed by a fixed-point equation as
(24)$$\gamma_{k,th}^{\left(n+1\right)}=e^{\frac{\ln2D_{k}}{m_{k}^{\left(t+1\right)}}+\sqrt{\frac{\ln2\sqrt{1-{\left(1+\gamma_{k,th}^{\left(n\right)}\right)}^{-2}}}{m_{k}^{\left(t+1\right)}}}\cdot Q^{-1}\left(\epsilon_{max}\right)}-1,\forall k\in K.$$

As $\gamma_{k}$ is still non-convex with respect to θ, the SINR constraint (Equation 23) is further transformed according to the following proposition.

**Proposition** **2.**
*At the t-th iteration, (Equation 23) is equivalent to*

(25)
$$\begin{array}{c} \theta^{H}R_{k}^{\left(t\right)}\theta +2R\left\{s_{k}^{\left(t\right)}\theta \right\}+v_{k}^{\left(t\right)}\leq 0. \end{array}$$

*where*

(26)
$$\begin{array}{c} R_{k}^{\left(t\right)}=\sum_{i\neq k}^{K}P_{i}Q_{i}^{H}u_{k}^{\left(t\right)}u_{k}^{(t)H}Q_{i},\;\;\;\;\;\;\;\;\;\;\;\;\;\;\;\;\;\;\;\;\;\;\;\;\;\;\;\;\;\;\;\;\;\;\;\;\;\;\;\;\;\;\;\;\;\;\;\;\;\;\;\;\;\;\;\;\;\;\;\; \end{array}$$


(27)
$$\begin{array}{cc} \; & s_{k}^{\left(t\right)}=\sum_{i\neq k}^{K}P_{i}u_{k}^{(t)H}h_{i}^{IC}Q_{i}^{H}u_{k}^{\left(t\right)}-\sqrt{P_{k}}Q_{k}^{H}u_{k}^{\left(t\right)},\\ \; & v_{k}^{\left(t\right)}=\sum_{i\neq k}^{K}P_{i}|{u_{k}^{(t)H}h_{i}^{IC}|}^{2}+\sigma^{2}{∥u_{k}^{\left(t\right)}∥}^{2}-2R\left\{\sqrt{P_{k}}u_{k}^{(t)H}h_{k}^{IC}\right\}+\gamma_{k,th}, \end{array}$$

*where*

(28)
$$\begin{array}{c} u_{k}^{\left(t\right)}=\sqrt{P_{k}}{\left(\sigma^{2}I_{N}+\sum_{i\neq k}^{K}P_{i}h_{i}^{\left(t\right)}h_{i}^{(t)H}\right)}^{-1}h_{k}^{\left(t\right)}. \end{array}$$



**Proof.** See Appendix A.    □

According to Proposition 2, ($P_{\theta }$) can be reformulated as
(29)$$\begin{array}{cc} (P_{\dot{\theta }}): & \underset{\theta }{\mathrm{minimize}}{∥\theta -\varphi^{\left(t\right)}+\lambda^{\left(t\right)}∥}^{2}\\ s.t.\; & \theta^{H}R_{k}^{\left(t\right)}\theta +2R\left\{s_{k}^{\left(t\right)}\theta \right\}+v_{k}^{\left(t\right)}\leq 0. \end{array}$$

The problem ($P_{\dot{\theta }}$) is a convex QCQP problem, and it can be easily solved through the existing solver such as CVX.

#### 3.2.4. Update φ

After obtaining the $m^{\left(t+1\right)},w_{k}^{\left(t+1\right)},\theta^{\left(t+1\right)}$ and $\lambda^{\left(t\right)}$, the auxiliary variable φ can be optimized by solving
(30)$$\begin{array}{cc} (P_{\varphi }): & \underset{\varphi }{\mathrm{minimize}}{∥\theta^{\left(t+1\right)}-\varphi +\lambda^{\left(t\right)}∥}^{2}\\ s.t.\; & |\varphi_{m}|=1. \end{array}$$

A closed-form solution of φ can be obtained as
(31)$$\varphi =e^{j∠(\theta^{\left(t+1\right)}+\lambda^{\left(t\right)})}$$

#### 3.2.5. Update λ

With the given $m^{\left(t+1\right)},w_{k}^{\left(t+1\right)},\theta^{\left(t+1\right)}$ and $\varphi^{\left(t+1\right)}$, the dual variable λ can be updated by
(32)$$\lambda^{\left(t+1\right)}=\lambda^{\left(t\right)}+\theta^{\left(t+1\right)}-\varphi^{\left(t+1\right)}.$$

Based on the above analysis, the overall algorithm to solve (P1) is summarized as Algorithm 1. The stopping criterion of Algorithm 1 is set to be ${∥\theta -\varphi ∥}^{2}<\varepsilon$.
**Algorithm 1:** The Proposed ADMM Algorithm.**Initialization:**$\left\{w_{k}^{\left(0\right)}\right\}$, $\theta^{\left(0\right)}$, $\lambda^{\left(0\right)}$, $\varphi^{\left(0\right)}$, iteration index *t* = 0.  **while** the stopping criterion is not met **do**     1: Update the blocklength $m^{\left(t+1\right)}$ by solving ($P_{m}$)     2: Update the receiving beamforming $w_{k}^{\left(t+1\right)}$ according to (Equation 21)     3: Update the variables $u_{k}^{\left(t+1\right)}$ according to (Equation 28)     4: Update the RIS phase shift $\theta^{\left(t+1\right)}$ by solving ($P_{\dot{\theta }}$)     5: Update $\varphi^{\left(t+1\right)}$ by solving ($P_{\varphi }$)     6: Update dual variables $\lambda^{\left(t+1\right)}$ by (Equation 32)     7: Update iteration index *t* = *t*+1  **end****Output:** $m^{*}$, $w_{k}^{*}$ and $\theta^{*}$.

### 3.3. Conventional SDR Method

In this subsection, the SDR-based RIS phase shift optimization will be introduced. Similar to our ADMM-based method, the optimal blocklength and received beamforming vector can be obtained by (Equation 19) and (Equation 21), respectively. For the given blocklength and received beamforming vector, (P1) can be reduced to a feasibility-check problem [10] as
(33)$$\begin{array}{cc} \mathrm{Find} & \;\theta \\ s.t.\; & \gamma_{k}\geq \gamma_{k,th}. \end{array}$$

Let $\bar{\theta }={\left[\theta ,1\right]}^{H}$ and define $V={\bar{\theta \theta }}^{H}$. We obviously have V⪰0, rank(V)=1 and $V_{m,m}=1$, where m=1,…,M+1. (Equation 33) can be then written as
(34)$$P_{k}\left[\mathrm{Tr}\left(Z_{k,k}\cdot V\right)+v_{k,k}^{2}\right]\geq \gamma_{k,th}\cdot \left\{\sum_{i=1,i\neq k}^{K}P_{i}\left[\mathrm{Tr}\left(Z_{k,i}\cdot V\right)+v_{k,i}^{2}\right]+\sigma^{2}\right\},$$
where $v_{k,i}=w_{k}^{H}\cdot h_{i}^{IC}$, $r_{k,i}=\mathrm{diag}\left(f_{i}^{H}\right)\cdot G\cdot w_{k}$, $Z_{k,i}=\left[\begin{array}{cc} r_{k,i}\cdot r_{k,i}^{H} & r_{k,i}\cdot v_{k,i}\\ r_{k,i}^{H}\cdot v_{k,i}^{H} & 0 \end{array}\right]$. As the rank-one constraint of V is non-convex, we formulate a convex problem by omitting the rank-one constraint as
$$(P_{V}):\mathrm{Find}\;V\;\;\;\;\;\;\;\;\;\;\;\;\;\;\;\;\;\;\;\;\;\;\;\;\;\;\;\;\;\;\;\;\;\;\;\;\;\;\;\;\;\;\;\;\;\;\;\;\;\;\;\;\;\;\;\;\;\;\;\;\;\;\;\;\;\;\;\;\;\;\;\;\;\;\;\;\;\;\;\;\;\;\;\;\;\;\;\;\;\;\;\;\;\;\;\;\;\;\;\;\;\;\;\;\;\;\;\;\;\;$$
(35)$$s.t.\;P_{k}\left[\mathrm{Tr}\left(Z_{k,k}\cdot V\right)+v_{k,k}^{2}\right]\geq \gamma_{k,th}\cdot \left\{\sum_{i=1,i\neq k}^{K}P_{i}\left[\mathrm{Tr}\left(Z_{k,i}\cdot V\right)+v_{k,i}^{2}\right]+\sigma^{2}\right\},$$
(36)V⪰0,                                                                                                         
(37)$$V_{m,m}=1,m=1,\ldots ,M+1.\;\;\;\;\;\;\;\;\;\;\;\;\;\;\;\;\;\;\;\;\;\;\;\;\;\;\;\;\;\;\;\;\;\;\;\;\;\;\;\;\;\;\;\;\;\;\;\;\;\;\;\;\;\;\;\;\;\;\;\;\;\;\;\;\;\;$$

By noting that ($P_{V}$) is a convex semi-definite program (SDP), it can be optimally solved by convex optimization solvers such as CVX. While the optimal V in ($P_{V}$) cannot be guaranteed to be a rank-one matrix, the Gaussian randomization technique is further applied to find a feasible solution from the obtained higher-rank solution of V.

### 3.4. Convergence and Time Complexity Analysis

Algorithm 1 iteratively minimizes the total blocklength of the considered system by alternatively updating blocklength vector m, the receive beamforming of the CC and the phase shift of RIS until convergence. Specifically, for any configuration of the receive beamforming vector of the CC and the RIS phase shift θ, the blocklength vector m can be analytically solved through a quadratic equation as (Equation 18). Given the blocklength m and RIS phase shifts θ, a Minimum Mean Square Error (MMSE) receiver is adopted to maximize the SINR of each IoT device. With the optimal blocklength vector and receive beamforming vector of the CC, the convex QCQP problem can be solved to optimize the RIS phase shift. By running such alternative optimization steps, the objective function in (P1) decreases monotonically. On the other hand, with a DEP limit for each IoT device, the blocklength would never converge to a trivial solution. Therefore, the convergence of Algorithm 1 can be guaranteed.

The time complexity to obtain blocklength vector m is O(K). The time complexity in optimizing the receive beamforming vector of the CC mainly lies on computing the inverse of *K* matrices, which is $O(KN^{3})$. The time complexity of solving the QCQP problem is $O(M^{3})$. Therefore, the overall complexity of the proposed ADMM algorithm is $O\{\max\left(KN^{3},M^{3}\right)\}$.

For the SDR-based method, the time complexities in solving m and $\left\{w_{k}\right\}$ are the same as those for the ADMM-based method. The time complexity in solving the RIS phase shift is $O({\left(M+1\right)}^{4.5}\log\left(\frac{1}{\varepsilon_{s}}\right))$ [20], where $\varepsilon_{s}$ denotes the accuracy of the SDR solution. Therefore, the overall time complexity of the SDR-based method is $O\{\max\left(KN^{3},{\left(M+1\right)}^{4.5}\log\left(\frac{1}{\varepsilon_{s}}\right)\right)\}$.

The time complexity of the SDR-based method should be mainly attributed to calculating a matrix with dimension M+1 through the interior point method and the Gaussian randomization technique to approximate a rank-one solution. For the ADMM-based method, the time complexity is dominated by solving the QCQP where a vector with dimension *M* should be optimized through the interior method.

## 4. Numerical Results

In this section, simulation results are presented to demonstrate the performance of our proposed design for the RIS-aided URLLC systems. In the simulation, we consider a Cartesian coordinate system, where a CC is located at (0, 0, 0), an RIS is deployed at (100 m, 10 m, 0 m), and *K* IoT devices are randomly located in a circle with radius of 5 m and center of (100 m, 0 m, 0m). ξ is set to be 3 dB under at the reference distance $d_{0}=1$ m. The path-loss exponents for the IoT-CC, the RIS-CC, and the IoT-RIS links are set to be 3.5, 2.5 and 2.0, respectively. Suppose that the packet lengths of all IoT devices are equal to *D* bit. The transmission power of each IoT device is set as *P*=10 dBm, and the noise power is set to be −70 dBm. The stopping criterion ε for convergence in Algorithm 1 are set as $10^{-4}$. The simulation results are averaged over 500 independent channel realizations.

Figure 2 presents how the total blocklength $\sum_{k=1}^{K}m_{k}$ varies with the maximum DEP under different algorithm strategies. For the sake of comparison, the performance without RIS and that solved by the SDR method is also plotted. As we can see from Figure 2, with RIS, the total blocklength significantly reduces compared to that without RIS. It can also be observed that the total blocklength reduces as the DEP increases. By noting that $m_{k}=T_{k}W_{k}$, where $T_{k}$ and $W_{k}$ are the transmission duration and the signal bandwidth of the *k*-th device, respectively, the blocklength $m_{k}$ is closely related to the transmission duration, which implies a trade-off between the latency and the reliability in the URLLC system. We can also observe from Figure 2 that our proposed ADMM-based method achieves a shorter total blocklength than the SDR-based method. When the number *M* of the reflecting elements on the RIS is large, the time complexity of our proposed ADMM-based method, i.e., $O\left\{\max(KN^{3},M^{3})\right\}$ is much lower than that of the SDR-based method, i.e., $O({\left(M+1\right)}^{4.5}\log\left(\frac{1}{\varepsilon_{s}}\right))$. We can then conclude that our proposed method achieves better performance than the conventional SDR-based method with much lower complexity.

Figure 3 presents how the total blocklength varies with the packet length *D*. As Figure 3 shows, the total blocklength with and without RIS both increase linearly with *D*. The gap between that with and without RIS increases when *D* becomes larger, which highlights the effectiveness of deploying RIS in the URLLC systems. When *D* is large, the blocklength $\left\{m_{k}\right\}$ has to be larger to satisfy the DEP constraint, leading to longer delay. It implies that to satisfy the latency and throughput requirement, we can conclude a proper choice of packet length, which also plays a vital role in our system.

Figure 4 shows how the total blocklength varies with the number of reflecting elements. As Figure 4 shows, with RIS, the total blocklength decreases as the number *M* of the reflecting elements increases. When *M* is large, our proposed design is able to significantly reduce the blocklength and thus reduce the latency. It indicates that it is necessary to deploy an RIS with a large number of reflecting elements to improve the URLLC system performance.

Figure 5 plots how the average blocklength, i.e., $\bar{m}=\frac{1}{K}\sum_{k=1}^{K}m_{k}$, varies with the number of devices *K*. As Figure 5 shows, the average blocklength $\bar{m}$ increases with *K*, as more devices are competing for the fixed amount of communication resources. Note that the gap between that with and without RIS becomes larger with *K*, which further implies the importance of deploying RIS under the strict QoS constraint in the URLLC systems.

Figure 6 further plots how the average blocklength varies with the number *N* of the antennas at the CC under different numbers of devices. The average blocklength $\bar{m}$ significantly decreases with *N*. The gap between that with different numbers of devices also reduces as *N* increases. Intuitively, when the number of CC antennas *N* increases, the dimension of the cascaded channel $h_{k}$ also increases, which provides more degrees of freedom for resource allocation. It indicates that to support the URLLC requirements, a large antenna array is also preferred at the CC even when the RIS is deployed.

## 5. Conclusions

In this paper, the uplink transmission of an RIS-assisted URLLC system is studied. To minimize the transmission latency under the reliability constraints, an optimization problem is proposed to jointly optimize the receiver, the channel blocklength and the RIS phase shifts. As the feasible region of the formulated problem is non-convex, a low-complexity algorithm is proposed based on the ADMM method. Simulation results verify that our proposed method is able to achieve better performance than the conventional SDR method with much lower complexity. The importance in deploying RIS in the URLLC systems is also discussed, which indicates that deploying RIS can drastically reduce the transmission latency, and the gain further increases with the number of reflecting elements at the RIS.

## Figures and Tables

**Figure 1 entropy-25-00857-f001:**
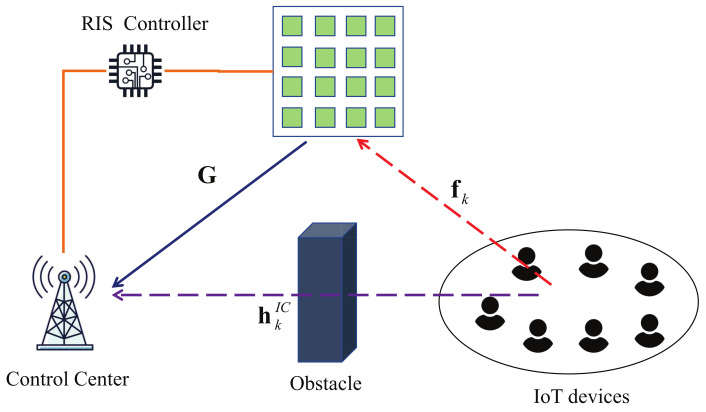
Architecture of our proposed reconfigurable intelligent surface (RIS)-aided ultra-reliable and low-latency communication (URLLC) system with a control center (CC), several IoT devices, and an RIS. An obstacle exists on the path between the CC and the IoT devices, and the RIS provides a reflection path to improve the channel quality.

**Figure 2 entropy-25-00857-f002:**
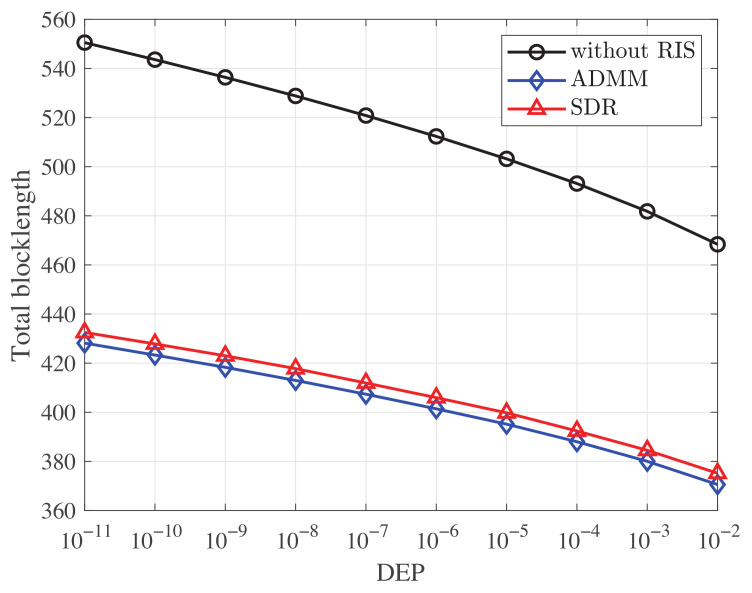
Total blocklength versus the decoding error probability (DEP) under various schemes. *M* = 20, *D* = 256 bits, *N* = 20, *K* = 10.

**Figure 3 entropy-25-00857-f003:**
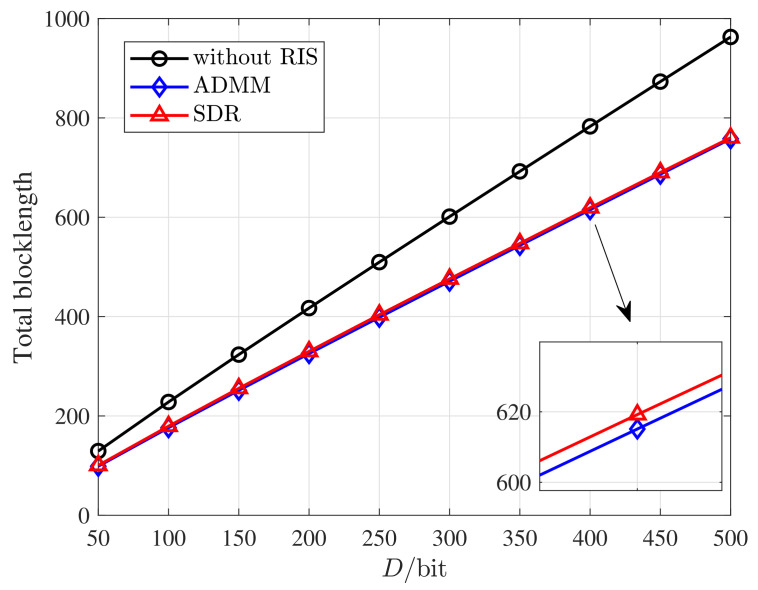
Total blocklength versus the packet length *D* under various schemes. *M* = 20, *N* = 20, DEP = $10^{-7}$, *K* = 10.

**Figure 4 entropy-25-00857-f004:**
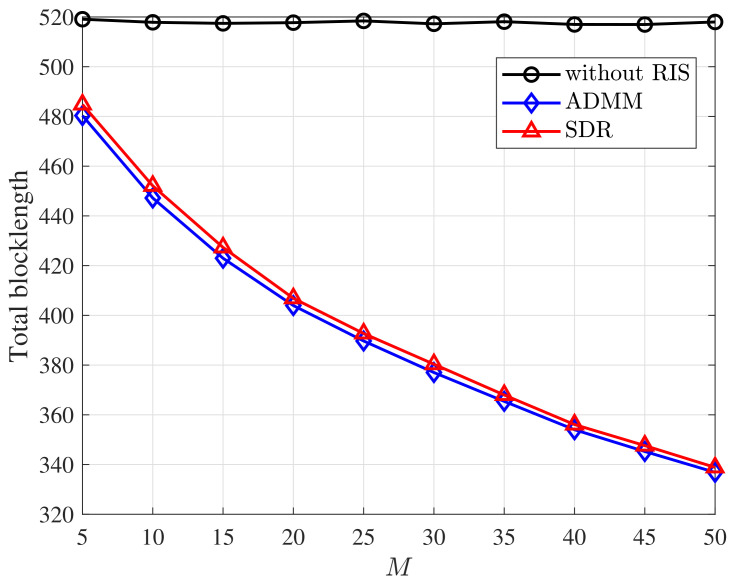
Total blocklength versus the number *M* of reflecting elements under various schemes. *D* = 256 bits, *N* = 20, *K* = 10, DEP = $10^{-7}$.

**Figure 5 entropy-25-00857-f005:**
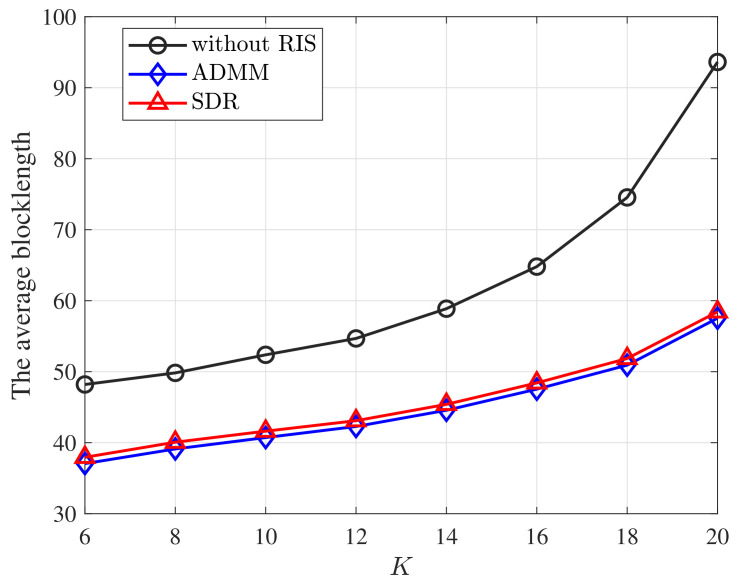
The average blocklength versus the number of device *K* under various schemes. *M* = 20, *D* = 256 bits, *N* = 20, DEP = $10^{-7}$.

**Figure 6 entropy-25-00857-f006:**
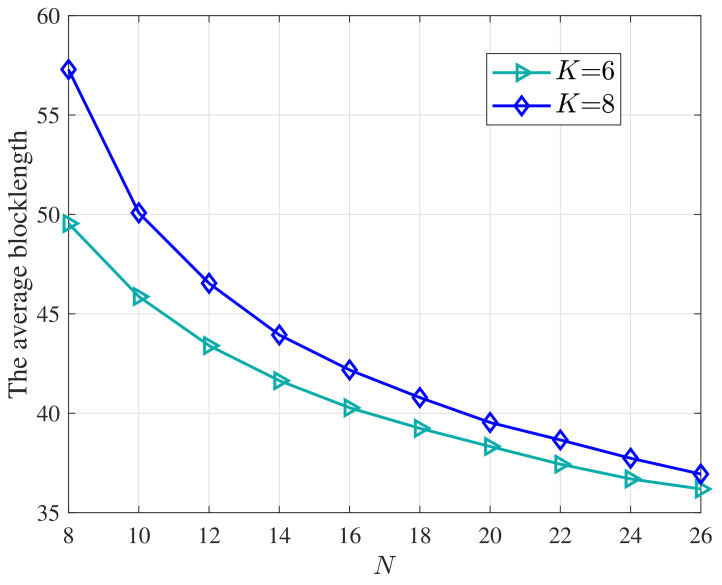
The average blocklength versus the number of CC antennas *N* under different numbers of IoT devices. *D* = 256 bits, *M* = 20, DEP = $10^{-7}$.

## Data Availability

Not applicable.

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
