# Peer review of "Delay Minimization in RIS-Assisted URLLC Systems under Reliability Constraints"

_entropy, 2023, doi:10.3390/e25060857_

Round 1
Reviewer 1 Report
This paper is very interesting and tries to minimize the delay in 5G ultra-reliable and low-latency communication (URLLC) under reliability constraints, assisted with reconfigurable intelligent surface (RIS).
Notice that RIS technology is quite new and it has been proposed to increase the system spectral and energy efficiencies by inducing independent phase shifts on the incident signals in the transmissions, in a similar way to beamforming techniques.
The formulated problem is non-convex, thus it is used a proposed low-complexity algorithm by using the Alternating Direction Method of Multipliers (ADMM) technique. By simulation it is shown that the proposed algorithm is able to achieve better performance than the conventional semi-definite relaxation (SDR) based.
The paper is interesting and opens the door to use this technology to improve URLLC communications.
The wording is clear. Nevertheless, the authors can improve this paper taking into account the following comments.
In section 2, Figure 1 it should be nice to introduce the abbreviations in this figure, caption and text to clarify it.
It should be interesting to review eq 2 and its explanation in the text.
In general, it should be better to use /cdot in the formulation for easy reading.
Also, it is necessary to introduce better eq 10 and P1 definition. Also, in the same way, eq 11 and 12 should require improvements for easy reading.
In Algorithm 1, it is better to explain how the inputs to the algorithm are provided.
Section 4, line 192 try to justify the large-scale fading of the links.
In the introduction, it should be nice to include some other references related to the proposed design for motivation, although with a different perspective such as “feasibility of a stochastic collaborative beamforming for long range communications in wireless sensor networks” in electronics.
Also, it is worth mentioning in the paper several recent applications where these techniques could be applied and fits well such as:
1.-spatio-temporal analysis of urban acoustic environments with binaural psycho-acoustical considerations for IoT-based applications” in sensors
2.- 5G IoT system for real-time psycho-acoustic soundscape monitoring in smart cities with dynamic computational offloading to the Edge," in ieee internet of things
In general, the paper is well structured and it can be enhanced following the previous suggestions.
Reviewer 2 Report
1. Introduce all abbreviation the first time they appear in the text (eg CC line51)
2. In figure 3 and 5 captions correct verus with versus.
3. In conclusion section you stated that Simulation results verify that our proposed method is able to achieve better performance than the conventional SDR method with much lower complexity.
This should be completed with supporting data in the manuscript as follows: a. First, a section describing the conventional SDR algorithm must be added as in the experimental results this algorithm is compared to the proposed one. b. Referring to the computational complexity, you should add a method for the evaluation of metrics designed to measure the computational complexity (time, space, communication bit complexity). The computational complexity should be evaluated for both compared algorithms (SDR and ADMM based proposed algorithm and results must be discuused). c. In the results section provide a discussion referring to the slight differences between the two methods (SDR and ADMM) at the cost of the calculated computational complexity.Author Response
Please see the attachment.

Round 2
Reviewer 1 Report
Now the paper has improved and it is easier for reading.